# Training Spatial-Frequency Visual Prompts and Probabilistic Clusters for Accurate Black-Box Transfer Learning

## ABSTRACT

Despite the growing prevalence of black-box pre-trained models (PTMs) such as prediction API services and proprietary software, there remains a significant challenge in directly applying general models to real-world scenarios due to the data distribution gap. Considering a data deficiency and constrained computational resource scenario, this paper proposes a novel parameter-efficient transfer learning framework for vision recognition models in the black-box setting. Our framework incorporates two novel training techniques. First, we align the *input space* (*i.e.*, image) of PTMs to the target data distribution by generating visual prompts of spatial and frequency domain. Along with the novel spatial-frequency hybrid visual prompter, we design a novel training technique based on probabilistic clusters, which can enhance class separation in the *output space* (*i.e.*, prediction probabilities). In experiments, our model demonstrates superior performance in a few-shot transfer learning setting across extensive visual recognition datasets, surpassing state-of-the-art baselines. Additionally, the proposed method efficiently reduces computational costs for training and inference phases. The code will be available upon publication.

## CCS CONCEPTS

• **Computing methodologies** → **Artificial intelligence**.

## KEYWORDS

Black-box pre-trained models, Transfer learning, Parameter-efficient fine-tuning, Visual recognition, Visual prompt tuning

## 1 INTRODUCTION

In recent years, machine learning (ML) models pre-trained with large-scale data have shown a significant performance improvement in visual recognition, even in zero-shot classification [3, 21, 22]. However, potential inheriting bias from their training datasets [6] make it difficult to properly generalize a distinct data distribution, necessitating a development of *domain-specific* ML models. Although transfer learning is a widely-investigated technique to re-purpose pre-trained models (PTMs) by transferring the knowledge learnt in a source-domain task to a target domain, it is expansive to obtain well-organized target-domain datasets (*e.g.*, medical datasets). In such data-deficient scenarios, parameter-efficient transfer learning (PETL) [8, 24, 36, 38], which aims to adapting classifier

**Unpublished working draft. Not for distribution.**

to downstream tasks by tuning only a small number of additional parameters, can be an economic and effective solution.

In addition to the challenge of dealing with data deficiency, the implementation of transfer learning methods in real-world industrial scenarios poses significant challenges. After great success of ChatGPT[1], there has been a noticeable trend towards the release of access-restricted PTMs in the form of APIs or software packages. Essentially, the intricate details of recent enterprise-level ML models, including their architecture, weights, training data, and techniques, are often kept confidential. While such API services offer efficient solutions for users with limited resources—such as small companies or individuals who face computational limitations when computing gradients through large-scale ML models—it remains challenging to adapt these opaque, *black-box* PTMs to domain-specific tasks that require tailored applications.

To address this challenge, we propose a novel tuning methodology for visual recognition models that adheres to the fundamental principles of operating within a black-box environment. In this setting, the method utilizes only the available input-output responses (given input data samples and their corresponding prediction probabilities) for tuning. Given that *the input-output response represents the only accessible information*, one may posit that strategically optimizing the input space can be an effective manner for enhancing model performance within such constraints. Although it may be feasible to tune the output space in white-box scenarios (*e.g.*, by applying a linear layer to the penultimate layer), these methods can become intractable in black-box settings. In this context, one can primarily focus on the concept of *visual prompting* [1, 30], which aims to learn pixel-level perturbation with the goal of adjusting the input space to better match the target distribution.

To date, studies focusing on visual prompting techniques in the black-box setting, where all network parameters and gradients are not directly accessible, are comparatively limited when contrasted with the white-box setting. Notably, only a few investigations, such as those documented in [20, 25], have delved into this area. The black-box adversarial reprogramming (BAR) method, introduced in [25], marks a pioneering effort in emphasizing the importance of input-space adaptation for black-box PTMs (PTMs). This method employs frame-shaped visual prompts (VPs) to effectively facilitate the adaptation process. In a subsequent development, BlackVIP [20] proposes the generation of input-dependent VPs through the use of an auxiliary encoder, which is specifically designed to extract features from images. This approach underscores the versatility and adaptability of visual prompting, significantly enhancing compatibility of models within specialized domains.

Building upon existing frameworks, our research aims to achieve two challenging objectives, *1) effective visual prompting* and *2) novel output-space tuning* strategies, respectively. For the first goal, we design a *spatial-frequency hybrid visual prompter*, where its primary

---

[1]https://openai.com/blog/chatgpt

motivation is to effectively manipulate images within the frequency domain. Techniques such as low-frequency perturbations, which are known to maintain the structural integrity of images, have demonstrated promising results in various studies related to domain adaptation [28, 31–33] and adversarial attacks [7, 13].

The efficacy of such frequency-domain manipulations, however, may vary across different datasets. To enhance the performance of frequency-domain VPs, our approach also integrates spatial-domain VPs. This integration is carefully managed to avoid conflicts between the two domains, thereby resulting in an effective adaptation strategy for PTMs within the input space across various target datasets. Our novel method of merging visual prompts from both spatial and frequency domains is accomplished through the creation of a spatial-frequency hybrid visual prompter. Notably, this integration is achieved without the reliance on an auxiliary encoder, illustrating the potential for improved adaptation by synergistically utilizing both spatial and frequency information.

The motivation of our second goal, output-space tuning, stems from the observation that prediction probabilities generated by PTMs frequently show minimal differences across classes when applied to a new domain. This similarity complicates the extraction of a distinct and robust learning signal, presenting a challenge for effective domain adaptation. In white-box PETL for PTMs, two strategies can be baseline approaches to address such problems: linear probing, which involves applying a learnable linear layer to the logits from the penultimate layer, and text prompt tuning for vision-language models. Nevertheless, the restrictive nature of the black-box setting makes it challenging to modify the output space directly. As a workaround, we propose trainable *cluster-based prediction refinement* as a strategy to enhance the output space that might otherwise be less distinguishable. This method specifically involves the utilization of auxiliary simplex prototypes to disentangle and reorganize the output space, making it more structured and easier to differentiate among classes.

In our experiments, we rigorously assess the performance of our model through few-shot transfer learning across a wide range of benchmark datasets, thereby showcasing the robustness and effectiveness of our proposed methodologies. Additionally, we conduct a thorough analysis of each component of our method to provide evidence supporting its architectural design. Our experiments further include an examination of computational overheads, specifically targeting memory efficiency and the reduction in training time, to highlight the practical advantages of our approach.

## 2 RELATED WORK

**Visual prompting.** The recent advent of foundation PTMs, as highlighted in key studies [3, 21, 22], has significantly increased interest in deploying these models for real-world applications. Despite their potential, a notable performance drop is often observed when these general-purpose models are applied directly to specialized domains. To bridge this gap, the concept of PETL [12, 24, 36, 38] has emerged as a prominent research direction. PETL focuses on the efficient fine-tuning of large models to tailor their capabilities more closely to specific domain requirements.

Visual prompting, one of promising approaches of PETL, aims to adapt the input data distribution for a specific target model

by introducing learnable, *dataset-specific perturbations at the pixel level* [2, 5]. Following this initial approach, subsequent studies have concentrated on refining the design of VPs. For example, Enhanced Visual Prompting [30] introduces a novel approach for integrating an image with a VP, creating a synergistic blend that enhances model performance. Additionally, Diversity-Aware Meta Visual Prompting [10] advocates for the development of multiple group-specific VPs. This method is designed to address the complex issue of distribution shift, aiming to more closely align with the original data distribution of the target model, thereby improving effectiveness across varied datasets.

Recently, a visual prompting in a black-box setting has been studied [20, 25] with an increasing attention towards privacy models. To the best of our knowledge, black-box adversarial reprogramming (BAR) [25] represents the initial effort to incorporate a VP into a black-box setting. This approach aims to effectively repurpose models pre-trained on natural images for applications in the medical domain. BlackVIP [20] propose an input-dependant VP design and the corresponding optimization method for black-box setting, and show a promising results on various benchmark datasets.

Unlike the previous methods, several studies [9, 18, 29] have ventured into examining the impact of VPs within the frequency domain. Initial investigations [18, 29] have shown that applying visual prompting techniques to the frequency domain can significantly enhance performance across a broad spectrum of downstream tasks. These tasks include unsupervised domain adaptation for segmentation, presenting the versatility and effectiveness of frequency-based visual prompting. Furthermore, Tsao et al. [26] have found the effect of harnessing low-frequency features to boost classification performance. This approach points towards the potential strategic importance of frequency components in visual prompts.

**Output space alignment.** In the context of general principles for black-box settings [20], the only resources available for training are each input sample along with its corresponding prediction probabilities. Consequently, there have been few attempts to tune the output space in such environments. In contrast, in white-box scenarios, a variety of techniques can be employed, such as adding a linear layer at the end of the network (*i.e.*, linear probing). Specifically, in the field of vision-language models [14–16, 21], there are numerous efforts to align the output space by tuning text embeddings.

Observations from research on text prompt tuning [35, 37, 38] serve as foundational inspirations for our cluster-based prediction refinement approach. Both CoOp [38] and CoCoOp [37] introduce methods for adjusting text prompts with learnable components, showcasing their effectiveness in achieving a more harmonious alignment between images and text. Moreover, earlier studies [35] highlight the critical role of text prompt tuning in alleviating low inter-class variance of text features, further underscoring the potential impact of such adjustments.

## 3 METHODS

**Problem statement.** Throughout this paper, we denote $\mathcal{D} \equiv \{(X_i, y_i)\}_{i=1}^{n}$ as a training set comprising $n$ data items, where $X_i \in \mathbb{R}^{h \times w \times c}$ is a $c$-channel 2D image of size $h \times w$, and $y_i \in \{1, \cdots, K\}$

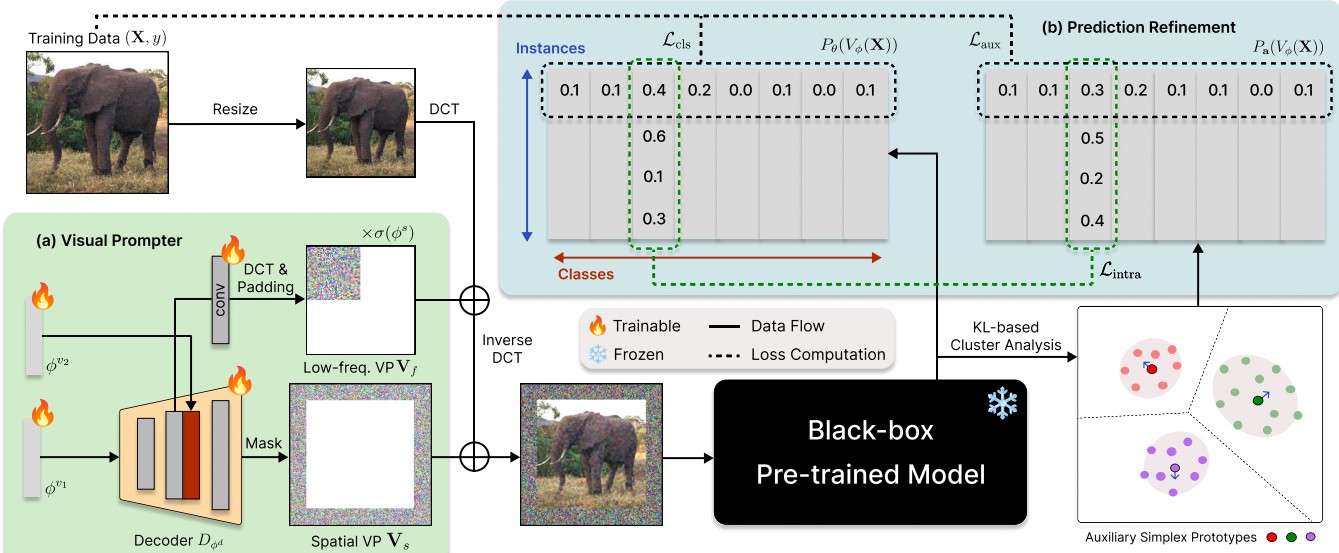

**Figure 1: A overall training workflow of our proposed method. (a) We illustrate our visual prompter consists of a single decoder $D_{\phi^d}$ and two trigger vectors ($\phi^{v_1}, \phi^{v_2}$), where the decoder simultaneously generates two VPs in spatial and frequency domains, respectively. Also, the learnable scaling parameter $\phi^s$ controls the effect of the low-frequency VP according to its efficacy. (b) After obtaining prediction probabilities $P_\theta(V_\phi(X))$, we conduct prediction refinement via KL-based cluster analysis. During training, we utilize auxiliary simplex prototypes to enhance the effectiveness of clustering based prediction refinement. After training the visual prompter based on $P_a(V_\phi(X))$, our method provides robust classification accuracy based on KL k-means refined probabilities $P_m(V_\phi(X))$ at inference time.**

is the corresponding class label. Given $\mathcal{D}$, this paper aims to construct a visual prompter $V_\phi$ to conduct PETL of a pre-trained visual recognition model $P_\theta$, as in previous works [2, 10, 30].

For our black-box problem formulation, we follow the black-box adversarial reprogramming (BAR) setting [25], which first defines the problem of black-box visual prompting. Specifically, it is assumed that $P_\theta$ is frozen and one can only access **1) each input sample** $(X_i, y_i)$ and **2) its prediction probabilities** $P_\theta(V_\phi(X))$. As it is intractable to compute gradient through $V_\phi$, the objective

$$\phi^* = \arg\min_\phi \mathbb{E}_{\mathcal{D}\sim(X,y)} \left[ \mathcal{L}(P_\theta(V_\phi(X)), y) \right]$$

can be optimized by zeroth-order optimization [17] to find an optimal $\phi$, where $\mathcal{L}$ is a task-specific loss function.

**Overview.** To further improve effectiveness when tuning a PTM in a black-box manner, we propose a novel visual prompter and first introduce an output-tuning method, where our contribution can be summarized as follows:

- For effective tuning of the input space of $P_\theta$ by using few-shot target data samples, we design a novel visual prompter $V_\phi$, which simultaneously provides prompts in the spatial and frequency domains. (Section 3.2)
- We propose a cluster-based class prediction strategy to flexibly manipulate the output probabilities of $P_\theta$. (Section 3.3)

- A number of training techniques are developed to further stabilize the training process of our visual-prompted black-box PTMs. (Section 3.4)

## 3.1 Preliminaries

In this subsection, we introduce preliminary concepts to facilitate a better understanding of our method.

**Visual prompting.** In BAR [25], its prompting method is defined by $V_\phi(X) = \text{Pad}(X_r) + V_\phi$, where $\text{Pad}(X_r)$ is a resized & zero-padded version of $X$ and $V_\phi \in \mathbb{R}^{h \times w \times c}$ is a VP (zero-masked at center) directly added to $X_r$, respectively. These frame-like VPs apply trainable pixel-level perturbations to the periphery of images. Various designs of frame VPs are possible; for instance, one might downsample an image and pad the VPs outside [25, 30], or overlap the VPs directly onto the original image [2]. Further techniques, such as those considering multiple subdomain-specific visual prompts (VPs) [10] and selecting VP sizes that vary according to dataset characteristics [27], have also been explored.

BlackVIP [20], a recently proposed black-box visual prompting method, addresses the challenges associated with the high parameter requirements and limited performance of BAR [25]. The approach effectively resolves these issues through the implementation of an encoder-decoder architecture in the visual prompter, which can be defined by

$$V_\phi(X) = X + \epsilon \cdot D_{\phi^d}(E(X) \oplus \phi^v). \tag{1}$$

In Eq. (1), $E$ is a self-supervised feature extractor, $D_{\phi^d}$ is a decoder generating a VP of size ($h \times w \times c$), $\phi^v$ is a trainable trigger vector, and $\epsilon \in (0, 1]$ is a hyperparameter.

**Zeroth-order optimization.** In the white-box scenario, where there is complete access to PTMs, optimizing the loss function and retrieving its gradient through back-propagation are straightforward. However, in the case of a black-box model, where only the model outputs are available, back-propagation is infeasible due to the unavailability of the gradient. In such situations, zeroth-order optimization emerges as a viable solution to this challenge. The two principal components of this method are 1) gradient estimation and 2) gradient descent using the estimated gradient.

In BlackVIP [20], the training parameters $\phi = \{\phi^d, \phi^v\}$ are optimized by the proposed SPSA-GC algorithm, which is a modified version of simultaneous perturbation stochastic approximation (SPSA) [23] based on the principle of Nesterov's accelerated gradient [19]. By using the conventional cross-entropy loss as $\mathcal{L}$, the authors conduct multi-point gradient estimation for SPSA, *i.e.*, the $t$-th iteration estimated gradient for the current parameters $\phi_t$ is

$$\hat{g}_t(\phi_t) = \frac{1}{S} \sum_{s=1}^{S} \frac{\mathcal{L}(\phi_t + c_t \Delta_s) - \mathcal{L}(\phi_t - c_t \Delta_s)}{2c_t} \Delta_s^{-1}, \quad (2)$$

where $c_i \in [0, 1]$ is a decaying parameter and $\Delta_s$ is a perturbation vector sampled from mean-zero distributions that satisfy finite inverse momentum condition such as Rademacher and segmented uniform distribution. For more in-depth description of the SPSA-GC optimization, readers are referred to the BlackVIP method [20].

**Discrete cosine transform.** The discrete cosine transform (DCT) plays a crucial role in converting spatial domain data—specifically, the actual pixels of an image—into the frequency domain. Each component of the DCT output represents a specific frequency present in the original image. The organization of these frequencies is such that low-frequency components are located in the top-left corner of the DCT matrix, while high-frequency components are positioned progressively towards the bottom-right. This structured arrangement enables a clear differentiation between varying types of image information, ranging from general trends to finer details. The ability to isolate and modify these frequencies is particularly advantageous, as it allows for targeted analysis and manipulation, thus providing insights into how different image characteristics influence the behavior of pre-trained models.

When performing 2D DCT on an $N \times N$-sized image, the operation can be defined by the following formula:

$$D(u, v) = \alpha(u)\alpha(v) \cdot$$
$$\sum_{x=0}^{N-1} \sum_{y=0}^{N-1} f(x, y) \cos\left[\frac{(2x+1)u\pi}{2N}\right] \cos\left[\frac{(2y+1)v\pi}{2N}\right], \quad (3)$$

where $D(u, v)$ is the DCT coefficient at frequencies $u$ and $v$, $f(x, y)$ is the pixel intensity at coordinates $(x, y)$, and $\alpha(u)$ and $\alpha(v)$ are normalization factors.

## 3.2 Spatial-Frequency Visual Prompter

Recall the visual prompter design as described in Eq. (1). This setup shows that an encoder-decoder-based VP generation can be a more effective solution than directly tuning pixel-level perturbations in resource-constrained black-box scenarios. However, as shown in its experiments, an inappropriate choice of encoder can lead to unintended outcomes and significant variability in effectiveness across different datasets. This is especially problematic given that the architecture of $P_\theta$ is not directly accessible. Moreover, $E$ incurs additional computational costs during training.

To alleviate the potential aforementioned issue and save additional computational costs (*e.g.*, memory and time) at training and inference phases, we design an *encoder-free* visual prompter. By keeping the decoder-based VP generation framework for efficiency, we aim to design a novel visual prompter capable of effectively adapting the input space of PTMs across various datasets. Inspired by the recent study [26], which investigated that manipulation in the low-frequency domain can facilitate efficient learning of VPs in certain datasets, we consider the following aspects:

- To enhance effectiveness, we incorporate a frequency-domain prompting strategy into our visual prompter.
- The impact of frequency-domain VPs can vary across datasets; therefore, we make their influence adjustable for each dataset and compensate with spatial-domain VPs.
- Considering a resource-constrained setting, we generate VPs in both the frequency and spatial domains in a memory-efficient manner, by using a single decoder structure.

As illustrated in Fig. 1(a). our visual prompter consists of a single decoder taking a learnable trigger vector $\phi^{v_1}$ as input. We use additional one $\phi^{v_2}$ to enable flexible generation of a spatial-domain VP. Denoting spatial and frequency VPs as $\mathbf{V}_s \in \mathbb{R}^{h \times w \times c}$ and $\mathbf{V}_f \in \mathbb{R}^{h_f \times w_f \times c}$, respectively, our decoder $D_{\phi^d}$ is

$$(\mathbf{V}_f, \mathbf{V}_s) = D_{\phi^d}(\phi^{v_1}, \phi^{v_2}), \quad (4)$$

where $h \gg h_f$ and $w \gg w_f$. Based on Eq. (4), our visual prompting process using $V_\phi$ can be described as follows:

**Frequency-domain prompting.** Image manipulation in the low-frequency band has become a popular strategy in adversarial attacks and adaptations [7, 13] to minimize redundant noise in the spatial domain and preserve the perceptual similarity of images. The low-frequency region, which accounts for most of an image's energy, is associated with content, whereas the high-frequency region pertains to edge and texture information.

Among the frequency analysis techniques such as fast Fourier transform and discrete wavelet transform, we employ discrete cosine transform (dct) and its inversion (idct) in our VPs. Given $\mathbf{X}$, downsample the image to $\mathbf{X}_r \in \mathbb{R}^{h_d \times w_d \times c}$, where $h_d > h_f$ and $w_d > w_f$. Then, frequency-domain prompted $\mathbf{X}_r$ is

$$\mathbf{X}_{r, fvp} = \text{idct}(\text{dct}(\mathbf{X}_r) + \sigma(\phi^s) \cdot \text{dct}(\mathbf{V}_f)), \quad (5)$$

where $\sigma$ and $\phi^s$ denote the sigmoid operation and a learnable scalar parameter for scaling the strength of frequency-domain prompting, respectively. In Eq. (5), we pad the bottom and right sides of the $\text{dct}(\mathbf{V}_f)$ with zeros to match it to the size of $\text{dct}(\mathbf{X}_r)$.

| Method | SVHN | Pets | EuroSAT | RESISC |
|---|---|---|---|---|
| Spatial (w/ Enc.) | 58.03 | 89.70 | 73.10 | 64.50 |
| Spatial (w/o Enc.) | 60.37 | 88.04 | 65.73 | 63.18 |
| Low freq. (w/ Enc.) | 66.57 | 88.55 | 69.04 | 61.67 |
| Low freq. (w/o Enc.) | 67.35 | 89.45 | 67.69 | 62.31 |

**Table 1: Exploratory results using BlackVIP (Spatial w/ Enc.) and its variants (Spatial w/o Enc., Low freq. w/ and w/o Enc.).**

**Spatial-domain prompting.**   After frequency-domain prompting, we apply spatial-domain prompting by

$$V_\phi(\mathbf{X}) = \mathsf{Pad}(\mathbf{X}_{r,fvp}) + \mathsf{Mask}(\mathbf{V}_s, \mathbf{X}_r), \tag{6}$$

where $\mathsf{Mask}(\mathbf{V}_s, \mathbf{X}_r)$ is an operation masking the center portion of $\mathbf{V}_s$ with zeros, matching the size of $\mathbf{X}_r$. In summary, our visual prompter $V_\phi$ applies both spatial- and frequency-domain VPs to each input image based on a single encoder $D_\phi^d$, two trigger vectors $(\phi^{v_1}, \phi^{v_2})$, and a scaling parameter $\phi^s$, *i.e.*, $\phi = \{\phi^d, \phi^{v_1}, \phi^{v_2}, \phi^s\}$.

## 3.3 Cluster-based Prediction Refinement

As mentioned, in the output space of a black-box visual recognition model, we can only obtain prediction probabilities. These probabilities tend to be computed using class-specific parameters that have been trained with source-domain data. For example, if logits are computed through the distances between class-specific prototypes and output features, each class prototype is dependent on the source-domain classes. Consequently, these prototypes may be inadequate for accurately representing target-domain classes. This inherent limitation persists even if the input space is manipulated to influence the stages just prior to the output space.

Furthermore, if the class prototypes remain fixed and are not adjusted to suit different domains, they maynot provide a strong learning signal to the input space. In situations where only probability vectors are accessible, we aim to enhance the flexibility of learning by redefining class-specific prototypes. We first explore the potential for prediction refinement through probabilistic clustering, and then define auxiliary simplex prototypes, Whose training techniques will be introduced in the next subsection.

**Probabilistic clustering for prediction refinement.**   Assuming that the largest index of $P_\theta(V_\phi(\mathbf{X}))$ can be a sub-optimal class prediction in some cases, one can use clustering methods for prediction refinement. Since only prediction probabilities are accessible, we apply KL k-means clustering [4] with $\{P_\theta(V_\phi(\mathbf{X}_i))\}_{i=1}^n$, which is a probabilistic clustering method for simplex data.

By clustering $\{P_\theta(V_\phi(\mathbf{X}_i))\}_{i=1}^n$ into $K$ clusters based on the KL divergence metric, where $K$ is the number of target classes, we obtain class-wise simplex means $\mathbf{m} = \{m_k\}_{k=1}^K$. Then, a refined version of $P_\theta(V_\phi(\mathbf{X}))$ can be formulated by

$$P_{\mathbf{m}}(V_\phi(\mathbf{X}))[k] = \frac{\exp(-\mathrm{KL}(P_\theta(V_\phi(\mathbf{X}))||m_k))}{\sum_{l=1}^K \exp(-\mathrm{KL}(P_\theta(V_\phi(\mathbf{X}))||m_l))}, \tag{7}$$

where $-\mathrm{KL}(P_\theta(V_\phi(\mathbf{X}))||m_k)$ works as a class-wise logit.

**Auxiliary simplex prototypes for training.**   The refinement strategy of Eq. (7) is a post-hoc method, *i.e.*, one can obtain informative class-wise simplex means $\{m_k\}_{k=1}^K$ after finishing the training process of $V_\phi$. To incorporate the principle into the training phase for a further enhancement, we introduce auxiliary simplex prototypes $\mathbf{a} = \{a_k\}_{k=1}^K$. In other words, one can use

$$P_{\mathbf{a}}(V_\phi(\mathbf{X}))[k] = \frac{\exp(-\mathrm{KL}(P_\theta(V_\phi(\mathbf{X}))||a_k))}{\sum_{l=1}^K \exp(-\mathrm{KL}(P_\theta(V_\phi(\mathbf{X}))||a_l))} \tag{8}$$

at training time to improve the class prediction accuracy of Eq. (7). To obtain initial prototypes $\mathbf{a}_0 = \{a_{0,k}\}_{k=1}^K$, we conduct clustering with initial classification prediction probabilities $\{P_\theta(\mathbf{X}_i)\}_{i=1}^n$. In the following section, we introduce our training strategies including the updating rules of the auxiliary simplex prototypes.

## 3.4 Training Probabilistic Clusters

In the previous subsection, we introduced the concept of auxiliary simplex prototypes. These prototypes enable us to fine-tune the output space of PTMs in conjunction with visual prompting at the input space. We anticipate that introducing flexibility in the output space may yield synergistic effects during VP training. Consequently, we have carefully designed training strategies that minimize conflicts between tuning in the input and output spaces. This consideration is crucial because we cannot process through PTMs in a straightforward manner; alternating updates between the input and output spaces separately may lead to unstable training.

**Objective function $\mathcal{L}$.**   When tuning a pre-trained visual recognition model, one can obtain $\{P_\theta(V_\phi(\mathbf{X}_i))\}_{i=1}^n$ and $\{P_{\mathbf{a}}(V_\phi(\mathbf{X}_i))\}_{i=1}^n$ with the training data $\{(\mathbf{X}_i, y_i)\}_{i=1}^n$, prediction probabilities of the black-box vision-language model and their refined version based on Eq. (8), respectively. Using the conventional cross-entropy loss function, we define $\mathcal{L}_{\mathrm{cls}} = \mathcal{L}_{\mathrm{CE}}(\{P_\theta(V_\phi(\mathbf{X}_i)), y_i\}_{i=1}^n)$ and $\mathcal{L}_{\mathrm{aux}} = \mathcal{L}_{\mathrm{CE}}(\{P_{\mathbf{a}}(V_\phi(\mathbf{X}))\}_{i=1}^n)$ for classification.

In addition to $\mathcal{L}_{\mathrm{cls}}$ and $\mathcal{L}_{\mathrm{aux}}$, we employ an additional loss function to enhance the alignment between the sets $\{P_\theta(V_\phi(\mathbf{X}_i))\}_{i=1}^n$ and $\{P_{\mathbf{a}}(V_\phi(\mathbf{X}_i))\}_{i=1}^n$. For this purpose, we adapt the intra-class relation loss [11], which is proposed to share the relational dynamics within each class in knowledge distillation. In our scenario, the relation loss $\mathcal{L}_{\mathrm{intra}}$ focuses on preserving the relative rankings or preferences of instances within a class, rather than replicating exact probability values from $\{P_\theta(V_\phi(\mathbf{X}_i))\}_{i=1}^n$ to $\{P_{\mathbf{a}}(V_\phi(\mathbf{X}_i))\}_{i=1}^n$. Unlike traditional methods that require strict output matching, $\mathcal{L}_{\mathrm{intra}}$ adopts a more relaxed approach, prioritizing the correlation of predictions. In summary, our overall loss function is defined by

$$\mathcal{L} = \mathcal{L}_{\mathrm{cls}} + \mathcal{L}_{\mathrm{aux}} + \mathcal{L}_{\mathrm{intra}}. \tag{9}$$

**Updating rules of a.**   Following the zeroth-order optimization method of the recently proposed black-box VP [20], we use the principle of the SPSA-GC algorithm to train our visual prompter $V_\phi$. Instead of updating $\mathbf{a}$ by computing gradients, which may not align with the gradients computed in the SPSA-GC algorithm for input-space VP training, we continuously update auxiliary simplex prototypes following the visual prompters' subsequent updates. Starting from the initial prototypes $\mathbf{a}_0 = \{a_{0,k}\}_{k=1}^K$, we update the

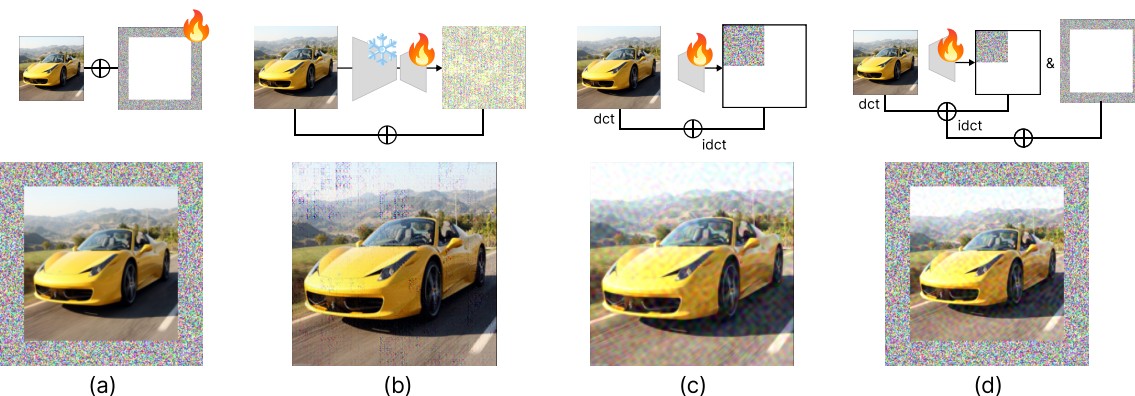

**Figure 2: An illustration of various visual prompting methods. (a) Visual prompting in the spatial domain is achieved by padding a VP outside each image. The VP itself can be trained using the principles of BAR [25]. (b) The spatial-domain visual prompting of BlackVIP [20], where VPs are generated by an encoder-decoder network. (c) Low-frequency visual prompting, where a decoder makes low-frequency visual prompts located in top-left corner (low-frequency in DCT). (d) Our Spatial-frequency visual prompting method, where spatial- and frequency-domain VPs are simultaneously constructed by a single decoder.**

prototypes by selectively obtaining the probabilistic distributions that result in a diminished loss direction.

To be specific, we discern between the outcomes of bidirectional perturbations ($\phi_t \pm c_t \Delta_s$ in Eq. (2)), and incorporate the probabilistic distribution that have lesser loss, thus having superior and accurate probabilistic representation. Averaging class-wise prediction probabilities, we refine the prototype vector in a weighted sum manner. Denoting $p_{t,k}$ as the average of the probability predictions for the $k$-th class training samples at the $t$-th iteration, we compute

$$a_{t+1,k} = \mathsf{L1}(0.9 \cdot a_{t,k} + 0.1 \cdot p_{t,k}), \qquad (10)$$

where L1 is $l_1$-normalization to maintain the probabilistic integrity of auxiliary simplex prototypes.

**Gradient surgery.** Although SPSA-GC has achieved faster convergence during training, the black-box optimization algorithm requires careful hyperparameter tuning to acquire desired performance. Noting that the multi-point gradient estimation of Eq. (2) simply averages $S$ estimated gradients to build a more stable gradient, we incorporate the principle of gradient surgery for robustness.

Inspired by a multi-task learning baseline [34], which project a conflicting gradient onto the normal plane of a gradient of other task, we seek to refine the estimation of a conflicting gradient.

Formally, let $\hat{g}_{t_1}, \hat{g}_{t_2}$ be a pair of sub-gradients among the the $S$ gradient estimations obtained concurrently at the same iteration step in Eq. (2). Then, the gradient projection process between two sub-gradients can be formulated by

$$\hat{g}_{t_1} = \begin{cases} \hat{g}_{t_1} - \frac{\hat{g}_{t_1} \cdot \hat{g}_{t_2}}{\|\hat{g}_{t_2}\|^2} \hat{g}_{t_2} & \text{if } \hat{g}_{t_1} \cdot \hat{g}_{t_2} < 0 \\ \hat{g}_{t_1}, & \text{otherwise.} \end{cases} \qquad (11)$$

We update each gradient estimation relative to other estimations to ensure that the revisions encompass all possible interactions between the gradients under consideration.

## 4 EXPERIMENTS

In this section, we describe the experimental settings (Section 4.1) including implementation details, datasets, and baselines. Before we present our main experimental results, we first introduce exploratory results related to our visual prompter (Section 4.2). Afterwards, we compare the performance of our model with baseline methods on benchmark and synthetic datasets (Section. 4.3), and then show performance improvement by each contribution (Section. 4.4).

## 4.1 Experimental Settings

**PTMs and Baselines.** As a representative pre-trained visual recognition model, we employ the CLIP zero-shot classifier [21], which can be extended across various numbers of classes and multi-modal scenarios, throughout our experiments. Specifically, along with each image X, CLIP typically takes a set of text prompts based on a hand-crafted template (*e.g.*, "a photo of a [CLASS].") The classifier then processes these prompts to extract text features, utilizing its tokenizer and text encoder. These extracted features are instrumental in generating class prediction probabilities.

For the baseline methods, we consider CLIP zero-shot classification (ZS), black-box adversarial reprogramming (BAR) [25], a black-box version of VP [1], and BlackVIP [20].

**Implementation details.** As aforementioned, we utilize CLIP VIT-B/16 [21] for a frozen PTM $P_\theta$, which is capable of strong zero-shot generalization. As the default input size of $P_\theta$ is ($224 \times 224 \times 3$), we resize each image into ($112 \times 112 \times 3$) or ($192 \times 192 \times 3$) according to its original resolution.

We adopt the decoder design from BlackVIP [20], in which low-frequency visual prompts (VPs) of size ($56 \times 56 \times 3$) are generated by applying a $1 \times 1$ convolution operation to its intermediate layer. For the scaling parameter $\phi^s$, we initialize it at $\phi^s = -5$ so that $\sigma(\phi^s) \approx 0$, thereby minimizing the impact of frequency-domain

| Method | Caltech | Pets | Cars | Flowers | Food | Aircraft | SUN | DTD | SVHN | EuroSAT | RESISC | CLEVR | UCF | IN | Avg. |
|--------|---------|------|------|---------|------|----------|-----|-----|------|---------|--------|-------|-----|-----|------|
| ZS | 92.9 | 89.1 | 65.2 | 71.3 | 86.1 | 24.8 | 62.6 | 44.7 | 18.1 | 47.9 | 57.8 | 14.5 | 66.8 | 66.7 | 57.6 |
| BAR | 93.8 | 88.6 | 63.0 | 71.2 | 84.5 | 24.5 | 62.4 | 47.0 | 42.7 | 77.2 | 65.3 | 18.7 | 64.2 | 64.6 | 61.4 |
| VP (black) | 89.4 | 87.1 | 56.6 | 67.0 | 80.4 | 23.8 | 61.2 | 44.5 | 61.3 | 70.9 | 61.3 | 25.8 | 64.6 | 62.3 | 61.2 |
| BlackVIP | 93.7 | **89.7** | 65.6 | 70.6 | 86.6 | 25.0 | 64.7 | 45.2 | 58.0 | 73.1 | 64.5 | 36.8 | 69.1 | 67.1 | 65.0 |
| **Ours** | **94.3** | 89.5 | **69.8** | **88.8** | **88.3** | **33.4** | **71.5** | **59.7** | **73.5** | **80.3** | **74.0** | **45.7** | **71.0** | **75.3** | **72.5** |

Table 2: The experimental results evaluating the classification performance of our framework and baseline methods in visual recognition tasks across 14 distinct benchmarks, including natural, specialized, structured, and fine-grained categories. Our approach is highlighted for its superior performance among black-box input-space prompting techniques. The experiments were conducted using a 16-shot learning framework, with each set of experiments repeated three times to ensure reliability.

prompting in the early training stages. Additionally, this parameter is designed to be adjustable as needed throughout training.

**Datasets.**  Following the protocol of BlackVIP [20], we use 14 image classification benchmark datasets and few-shot training & validation: 16-shot for training and 4-shot for validation. The datasets include a wide range of domains such as natural image, remote sensing, and textures, which can evaluate our framework on diverse modalities. For further analyses, we additionally employ two synthetic datasets: Biased MNIST and Loc-MNIST.

## 4.2 Exploratory Results

In this subsection, we outline two foundational observations that underpin the development of our visual prompter design. These observations are essential for guiding the architectural decisions and functional strategies we implement. Our aim is to craft a visual prompter that is not only effective but also aligns with our goals of enhancing model performance.

**Frequency-domain prompting.**  Recall the framework of Black-VIP [20], which makes spatial-domain VPs of the same size as each image based on an encoder-decoder structure. To design an effective visual prompter without encoder, we first construct three variants of BlackVIP (spatial-domain VP with an encoder) and test them with a number of datasets, SVHN, Oxford-Pets, EuroSAT, and RESISC. Here, the variants are 1) spatial-domain VP without encoder, 2) low-frequency VP with an encoder, and 3) low-frequency VP without encoder, where we follow the experimantal settings of BlackVIP. In Fig. 2, we illustrate various visual prompting methods for a better understanding of spatial- and frequency-domain prompting.

Table 1 illustrates that frequency-domain prompting can successfully adapt to certain datasets, such as SVHN, even without a self-supervised encoder, although it may yield suboptimal results in other contexts. To leverage the benefits of both spatial and frequency domain prompting without the drawbacks of each interfering, we have developed our visual prompter as shown in Fig. 1. In this design, we introduce a learnable scaling parameter, $\phi^s$, which enables $V_\phi$ to dynamically adjust the impact of low-frequency prompting. This allows for customized application, tailored to maximize effectiveness in various contexts.

**Trigger vectors.**  To enable effective tuning, we introduce a visual prompter designed to produce VPs in both the spatial and frequency domains. Our approach involves training a singular decoder that

is proficient in generating VPs across these two distinct domains simultaneously. This is achieved by employing two trigger vectors, where an additional trigger vector is incorporated into the decoder after extracting features for frequency-domain prompting. Such strategy of Eq. (4) is visually represented in Fig. 1(a).

We evaluated the impact of incorporating an additional trigger vector in dual-domain visual prompt generation by comparing training outcomes with and without it. Anticipating that the trigger vector would provide increased learning flexibility and enhance adaptability across different domains, our findings confirmed that its inclusion significantly enhanced both performance and the overall learning process compared to the version without it.

## 4.3 Experimental results

**Transfer learning on benchmark datasets.**  Utilizing 14 benchmark datasets, we implement a few-shot approach as described in previous studies [2, 37, 38] to conduct our evaluations, with the results detailed in Table 2. Our findings suggest that our method consistently outperforms existing black-box visual prompting techniques in terms of overall performance. Additionally, in certain cases, it achieves results that are comparable to those of white-box methods, thereby demonstrating the effectiveness and adaptability of our approach across diverse testing environments.

Our method has demonstrated considerable superiority across multiple datasets by substantial margins, underscoring the effectiveness of our visual prompter, even in the absence of an encoder integration. This enhanced performance is particularly evident when compared to various baseline variants, as detailed in Table 1. These comparisons effectively highlight the benefits of our hybrid approach, which combines spatial and frequency-domain prompting, thereby validating the design's efficacy in boosting model performance across diverse testing environments.

**Transfer learning on synthetic datasets.**  To evaluate the robustness of visual prompting methods against distribution shifts, we follow the experimental protocol outlined by Oh et al. [20], using datasets such as Biased MNIST [1] and Loc-MNIST. This approach allows us to systematically assess the effectiveness of these methods in varying data environments.

Biased MNIST is designed to examine generalization ability under color bias shift. At training, every digit is assigned a distinct background color that is closely linked to its class label, where the correlation strength is set by the value of $\rho \in [0, 1]$. In test phase, the correlation ratio is inverted to $1 - \rho$. Table 3 shows that our

| Methods | Biased MNIST | | | | Loc-MNIST | | | |
| --- | --- | --- | --- | --- | --- | --- | --- | --- |
| | 16-shot | | 32-shot | | 16-shot | | 32-shot | |
| | $\rho = 0.8$ | $\rho = 0.9$ | $\rho = 0.8$ | $\rho = 0.9$ | 1 : 1 | 1 : 4 | 1 : 1 | 1 : 4 |
| VP (white) | 57.92 | 43.55 | 69.65 | 42.91 | 86.79 | 86.54 | 90.18 | 92.09 |
| ZS | 37.56 | 37.25 | 37.56 | 37.25 | 29.70 | 22.70 | 29.70 | 22.70 |
| BAR | 53.25 | 53.07 | 53.93 | 53.30 | 33.98 | 26.05 | 34.73 | 27.72 |
| VP (black) | 60.34 | 53.86 | 59.58 | 51.88 | 16.21 | 25.68 | 18.43 | 30.13 |
| BlackVIP | 66.21 | 62.47 | 65.19 | 64.47 | **69.08** | 60.86 | 76.97 | 67.97 |
| **Ours** | **73.17** | **75.16** | **76.53** | **76.38** | 69.01 | **65.16** | **77.10** | **70.31** |

**Table 3: The experimental results evaluating the robustness against distribution shift of our framework and baseline methods. Using two synthetic datasets, Biased MNIST and Loc-MNIST, the experiments were conducted using 16- and 32-shot learning frameworks, with each set of experiments repeated three times to ensure reliability.**

| | Components | SVHN | EuroSAT | DTD |
| --- | --- | --- | --- | --- |
| (1) | Hybrid visual prompting | 70.1 | 75.0 | 48.5 |
| (2) | + Prediction refinement | 71.3 | 76.7 | 51.1 |
| (3) | + Auxiliary simplex training | 72.4 | 78.9 | 58.0 |
| (4) | + Intra-class relation loss | 73.0 | 79.6 | 58.9 |
| (5) | + Gradient surgery | 73.5 | 80.3 | 59.7 |

**Table 4: Ablation study results, where we incrementally integrated the proposed techniques: spatial-frequency visual prompting, prediction refinement, auxiliary simplex training, intra-class relation loss, and gradient surgery.**

approach yields better outcomes than the previous methods. These results may indicate that input-dependent prompting isn't always necessary for distribution shift robustness, as it can lead to adverse effects when unrelated information is added to images.

To further validation, Oh et al. [20] develop Loc-MNIST, which involves placing an actual target digit on one of the four edges and a random false digit in the middle of a blank image. Both the position of the target digit and the type of the fake digit are randomly selected. Additionally, a more complex scenario where the fake digit is four times bigger (1:4 ratio) than the real one is considered. We also present the corresponding results in Table 3, which complements the robustness analysis using Biased MNIST.

### 4.4 Ablation Study and Analysis

To assess the individual contributions of the components in our black-box visual prompting method, we conducted an ablation study. Beginning with the basic spatial-frequency visual prompting, we gradually incorporated the proposed techniques: prediction refinement, auxiliary simplex training, intra-class relation loss, and gradient surgery. This systematic approach helped us to meticulously analyze and comprehend the influence of each component on the overall performance.

Referring to Table 2, it is evident from Table 4 that our hybrid visual prompter achieves higher classification accuracy compared to existing state-of-the-art black-box tuning methods. Additionally, it is clear that employing prediction refinement through KL k-means clustering consistently enhances recognition accuracy. The auxiliary simplex training strategy further improves accuracy,

particularly in certain datasets (e.g., DTD). Furthermore, the intra-class relation loss and gradient surgery strategies have proven effective in stabilizing the training process.

**Computational costs.** In contrast to BlackVIP [20], a state-of-the-art black-box visual prompting technique that employs an encoder-decoder architecture, our approach achieves greater efficiency during training, due to the omission of the self-supervised encoder. Through experimental evaluation on the SVHN and EuroSAT datasets, utilizing an NVIDIA RTX A5000 GPU, our method showed a more resource-efficient footprint, consuming 2970MB of GPU memory and requiring only 0.225 seconds per iteration. This is in comparison to the previous method, which necessitated 4223MB and 0.374 seconds for the same metrics, respectively. This efficiency gain underscores the practical advantages of our methodology.

## 5 CONCLUSION

In conclusion, our work has successfully implemented a novel transfer learning framework for large PTMs that effectively bridges the data distribution gap in resource-constrained black-box API environments. By integrating spatial-frequency hybrid visual prompts with cluster-based prediction refinement, we have achieved significant improvements in few-shot transfer learning performance across a variety of datasets. This approach also results in reduced computational costs. Our methods enhance the adaptability and efficiency of PTMs in practical applications, while also fostering new research opportunities in transfer learning and model adaptation. The progress documented in this paper sets the stage for further investigation and advancement in the field, promising to enhance the utility and reach of machine learning across multiple scenarios.

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
