# OpenReview forum: "Training Spatial-Frequency Visual Prompts and Probabilistic Clusters for Accurate Black-Box Transfer Learning"
_acmmm.org/ACMMM/2024/Conference — MM2024 Poster_

### Official Review · Reviewer_navg · 2024-05-05

**Rating:** 4
**Confidence:** 2

**Summary:**

This manuscript introduces a novel transfer learning framework tailored for vision recognition models, particularly effective in black-box scenarios. The authors adeptly tackle the challenge of deploying generalized models into real-world contexts, where discrepancies in data distribution are prevalent. The proposed framework is distinguished by its integration of two innovative training methodologies: spatial-frequency hybrid visual prompting and probabilistic cluster-based prediction refinement. The empirical findings underscore the framework's exceptional performance in few-shot transfer learning environments and across a diverse array of visual recognition datasets. Furthermore, the authors highlight the framework's commendable efficiency, significantly reducing computational expenses during both the training and inference phases.

**Strengths:**

1. The paper presents an innovative, parameter-efficient transfer learning framework that leverages spatial-frequency hybrid visual prompts and probabilistic clusters, effectively addressing a critical issue in the field of visual recognition.

2. The methodology's incorporation of zeroth-order optimization and discrete cosine transform (DCT) showcases a significant technical advancement in the domain.

**Limitations:**

1. The Introduction section lacks the precision and clarity necessary to effectively convey the paper's research objectives and contributions, which may impede the reader's understanding and appreciation of the work's significance.

2. While the paper introduces a transfer learning framework that aims to be parameter-efficient, it falls short of clearly delineating how its spatial-frequency hybrid visual prompts and probabilistic clusters offer a significant advancement over existing methodologies in visual recognition.

3. The manuscript's failure to extend beyond the current paradigms of visual prompting and cluster-based prediction refinement is a notable weakness. The paper does not clearly articulate how the new training techniques contribute to the evolution of these frameworks.

4. A significant shortcoming of the manuscript is the absence of visual analysis techniques, such as Grad-CAM analysis and prompt visualization. These methods are indispensable for offering insights into the model's decision-making mechanisms and for understanding the importance of the framework's components. The lack of such visualizations detracts from the paper's ability to foster interpretability and transparency, which are crucial for establishing credibility and understanding among the research community and potential end-users.

5. A potential limitation of the proposed methods could be the complexity of implementation, especially for users without a deep understanding of transfer learning or visual recognition models. Ensuring clear documentation and support for code replication would be beneficial.

I will revise my ratings based on other comments and the author's rebuttals.

**Suitability:**

2

---

### Official Review · Reviewer_6Z9W · 2024-05-20

**Rating:** 4
**Confidence:** 2

**Summary:**

The paper introduces a novel parameter-efficient transfer learning framework for vision recognition models in a black-box setting. By aligning input space and generating visual prompts, along with utilizing probabilistic clusters, the model demonstrates superior performance in few-shot transfer learning scenarios. Experimental validation on extensive visual recognition datasets showcases the effectiveness of the proposed method.

**Strengths:**

(1) The paper introduces a novel transfer learning framework for large pre-trained models (PTMs) that addresses the data distribution gap in resource-constrained black-box API environments.
(2) Experimental results on synthetic datasets like Biased MNIST and Loc-MNIST, as well as benchmark datasets like SVHN, EuroSAT, and DTD, demonstrate the effectiveness of the hybrid visual prompting method and its enhancements.

**Limitations:**

(1)While the paper briefly mentions related work on visual prompting and transfer learning, a more comprehensive review and analysis of existing literature in the field would provide better context for the proposed framework. A deeper discussion of how the current work builds upon or differs from prior research would strengthen the paper's contribution.
(2)The experimental details are incomplete, with a lack of specific implementation details for training, making it difficult for other researchers to reproduce the findings presented in the paper.
(3)In addition to providing VIT-B/16 as the baseline performance, the paper should also include the performance of baseline models such as RN101 and ViT-B/32 to adequately demonstrate the effectiveness of the proposed method.
(4)Although the paper shows some comparative experiments comparing performance with and without encoders and in different domains, it lacks comparison with other state-of-the-art black-box transfer learning methods.
(5)The method's complexity and computational requirements are not thoroughly discussed, which could impact scalability. Furthermore, the authors should provide computational cost comparisons with other methods.

**Suitability:**

2

---

### Official Review · Reviewer_gZD6 · 2024-05-25

**Rating:** 4
**Confidence:** 2

**Summary:**

The paper introduces a new transfer learning framework for black-box pre-trained models (PTMs), aimed at addressing data distribution gaps and computational limitations. It uses spatial-frequency hybrid visual prompts to align PTMs' input space with the target data distribution and employs a probabilistic clustering method to enhance class separation in the output space. Evaluations on various visual recognition datasets demonstrate improved performance in few-shot transfer learning while reducing computational costs.

**Strengths:**

1. Enhancing black-box PTMs is a topic worthy of research.

2. The framework is computationally efficient, a critical factor for applications with constrained resources.

3. The paper includes extensive experiments on multiple datasets, demonstrating the robustness and effectiveness of the proposed method.

**Limitations:**

1. The spatial prompt and the frequency prompt are both generated from a single decoder. What is the impact on the final results when using two decoders instead? Will generating them from a single decoder cause any conflicts?

2. In line 572, "Updating rules of a," please add the definition of 'a.' This will enhance the readability of the paper.

3. In Eq. (10), is the selection of the hyper-parameters 0.9 and 0.1 sensitive?

4. Both [a] and [b] use visual prompts for transfer learning. Can the authors compare the impact of different visual prompt placements on the model's performance?

[a] Decorate the newcomers: Visual domain prompt for continual test time adaptation

[b] Exploring Sparse Visual Prompt for Domain Adaptive Dense Prediction

**Suitability:**

2

---

### Official Review · Reviewer_wrkc · 2024-05-26

**Rating:** 4
**Confidence:** 3

**Summary:**

This paper proposes a parameter-efficient transfer learning method, containing two training strategies: 1) aligning input space with target distribution by both spatial and frequency domains; 2) enhancing the class separation in the output space by probabilistic clustering. Experiments on standard benchmark show the performance of the proposed method.

**Strengths:**

1. The proposed method shows good average performance against other considered methods.

2. The paper is well-structured and easy to follow.

**Limitations:**

1. In Section 3.2, the paper argues the drawbacks of including encoders, thus choosing to only include decoders in the model. More justifications are needed for such a choice.

2. For 'updating rule of a', the choice on normalisation method and coefficient for weighted sum should also be analysed.

3. The setup solely considers the 16-shot learning framework and ViT-B/16 as the backbone. Additional experiments are required to show the validity and effectiveness of the method.

4. The novelty is somehow limited. The method is proposed by modifying existing methods with techniques that are used in other areas.

**Suitability:**

2

---

### Meta-Review · Senior_Area_Chairs · 2024-07-12

**Recommendation:** Accept (Poster)
**Confidence:** 4

**Metareview:**

this paper addresses the problem of parameter efficient transfer learning. The 4 reviewers appreciate the originality of the paper as well as the quality of the answers in the rebuttal material. Authors made there several promises which should be implemented in order to strongly enhance the quality of the manuscript -- the questions of reviewers were instrumental here. All reviewers judge that this contribution is too unimodal in nature, but it is understood that the contribution can be profitable to the MM  community.

the quality of the work, the settings and the originality of the approach push the recommendation to ACCEPT, poster.